# Does Accelerometry at the Centre of Mass Accurately Predict the Gait Energy Expenditure in Patients with Hemiparesis?

**DOI:** 10.3390/s23167177

**Published:** 2023-08-15

**Authors:** Léo Barassin, Didier Pradon, Nicolas Roche, Jean Slawinski

**Affiliations:** 1UMR 1179 END-ICAP, UVSQ, 78000 Versailles, France; leo.barassin@ispc-synergie.org (L.B.); didier.pradon@aphp.fr (D.P.); roche.nicolas@aphp.fr (N.R.); 2Pôle Parasport Santé, CHU Raymond Poincaré, APHP, 92380 Garches, France; 3ISPC Synergies, 75008 Paris, France; 4Service Explorations Fonctionnelles, CHU Raymond Poincaré, APHP, 92380 Garches, France; 5EA 7370 Laboratoire SEP, INSEP, 75012 Paris, France

**Keywords:** accelerometery, energy expenditure, stroke, gait, six-minute walking test

## Abstract

Background: The aim of this study was to compare energy expenditure (EE) predicted by accelerometery (EE_A_cc) with indirect calorimetry (EE_META_) in individuals with hemiparesis. Methods: Twenty-four participants (12 with stroke and 12 healthy controls) performed a six-minute walk test (6MWT) during which EE_META_ was measured using a portable indirect calorimetry system and EE_ACC_ was calculated using Bouten’s equation (1993) with data from a three-axis accelerometer positioned between L3 and L4. Results: The median EE_META_ was 9.85 [8.18;11.89] W·kg^−1^ in the stroke group and 5.0 [4.56;5.46] W·kg^−1^ in the control group. The median EE_ACC_ was 8.57 [7.86;11.24] W·kg^−1^ in the control group and 8.2 [7.05;9.56] W·kg^−1^ in the stroke group. The EE_ACC_ and EE_META_ were not significantly correlated in either the control (*p* = 0.8) or the stroke groups (*p* = 0.06). The Bland–Altman method showed a mean difference of 1.77 ± 3.65 W·kg^−1^ between the EE_ACC_ and EE_META_ in the stroke group and −2.08 ± 1.59 W·kg^−1^ in the controls. Conclusions: The accuracy of the predicted EE, based on the accelerometer and the equations proposed by Bouten et al., was low in individuals with hemiparesis and impaired gait. This combination (sensor and Bouten’s equation) is not yet suitable for use as a stand-alone measure in clinical practice for the evaluation of hemiparetic patients.

## 1. Introduction

Physical activity measurements have been used to indirectly quantify energy expenditure in individuals with various pathologies for several years [1,2,3,4]. Connected devices, such as watches, bracelets, and smartphone applications, which are designed to increase the activity levels of the general public, have become popular among clinicians due to their ease of use and low cost. Such devices have thus been integrated into clinical practice and research to indirectly quantify energy expenditure [5]. Studies comparing results from off-the-shelf connected devices with specialised, equivalent medical devices or indirect calorimetry (which is the gold standard) have found that they accurately record data, such as the number of steps and distance covered, and reliably estimate energy expenditure in healthy subjects [6,7]. However, the methods used to predict energy expenditure by Mandigout et al. [8], where the prediction was made using an accelerometer, have been protected as an industrial secret.

Increasing the level of physical activity for people with a chronic pathology, such as stroke, has been shown to reduce their co-morbidities [9,10]. The evaluation of the impact of stroke treatments would be improved if clinicians could reliably and easily measure the amount of activity performed by their patients [11]. A previous study has shown that patients with stroke are more inactive than healthy age-matched controls [12]. Research has also shown that energy expenditure is doubled in patients with stroke due to the sequalae (mainly weakness and spasticity) of their hemiparesis [13].

Feedback on patients’ activity levels would not only inform healthcare providers, but it might also motivate individuals with stroke to perform regular physical activity, and is therefore recommended by the HAS (Haute Autorité de Santé) [14]. The accessibility of new technologies and connected devices that are easily integrated into peoples’ daily lives and which allow activity to be tracked, such as smartphone applications and smart watches, have simplified the collection of detailed data relating to physical activity levels outwith the hospital setting [15]. Nevertheless, several studies have indicated that inter-device reliability can be poor due to factors including the device’s position on the body, the recording method used, and the equations used to process the data, all of which result in either an over- or underestimation of energy expenditure [16]. As a result, the use of connected devices is currently a less reliable measurement technique than indirect calorimetry [17].

Therefore, despite the promise of such devices, the clinical interest in them, and the work on their development, there is currently no consensus on their use in individuals with chronic diseases and significant gait asymmetry. Optimal sensor types and positions for the accurate evaluation of physical activity levels and energy expenditure have yet to be identified. One method frequently reported in the literature [18,19,20,21] is Bouten’s method [22]. This method has been validated in healthy individuals, but not in people with gait disorders [23]. Bouten’s method [22] uses a regression equation to calculate the integral of signal data recorded using an accelerometer, positioned between L3 and L4 (so as to be close to the person’s centre of mass) in three planes of space (x, y, and z) in order to estimate energy expenditure during the gait.

Moreover, following a cardiovascular accident (stroke), we often observe motor impairment caused by either a hemorrhage (hemorrhagic stroke) or a blocked artery (ischemic stroke) in the motor cortex. Neuromuscular disorders result from that, causing locomotor impairments. In terms of spatiotemporal parameters of the gait cycle, reductions in speed, cadence, and stride length have been observed [24,25]. At the joint kinematic level, disturbance in flexion has been observed [26]. At the hip level, there can be limitations in knee elevation due to impaired flexion and/or hip extension [27]. This can lead to difficulties in overcoming obstacles. In terms of the knee joint, during the stance phase, hyperextension and a deficit in flexion during the swing phase can be observed [28]. These issues can be explained, on one hand, by the overactivity of the triceps surae, resulting in knee extension and plantar flexion disturbance; on the other hand, it is possibly due to the overactivation of the rectus femoris. Finally, at the ankle level, there is often hyperactivity of the plantar flexors and weakness of the dorsiflexors. These impairments can lead to foot drop [29]. The aforementioned impairments result in a significant increase in energy cost during walking [25]. This means that the patient will expend more energy per unit of distance compared to someone without a pathology [30]. The need to evaluate the effects of therapies on these gait disorders is essential. Consequently, the evaluation of the energy cost of walking, or more simply of energy expenditure, is relevant to support clinicians in the overall evaluation of the effects of the therapies chosen. In fact, the connected objects allowing this indirect measurement have a preponderant place in the evaluation of the impact of therapeutics on the autonomy of walking. Thus, a question arises: are connected tools using the Bouten’s method sufficiently accurate to estimate energy expenditure in patients who have had a stroke?

The aim of this study, therefore, was to compare the accuracy of energy expenditure values calculated using Bouten’s regression equation method [22] with those obtained from the gold standard method of indirect calorimetry. This work would help to validate the use of Bouten’s method as a simple way to assess people who have stroke-related hemiparesis and impaired gait. Data were compared for both methods from two groups of subjects, n = 12 individuals with stroke and impaired gait, and n = 12 healthy controls during a six-minute walk test (6MWT).

## 2. Materials and Methods

### 2.1. Participants

Participants with stroke or impaired gait were recruited either during a routine follow-up medical consultation, or while they were hospitalised in rehabilitation. Inclusion criteria were as follows: aged over 18 years, able to walk without assistance or assistive devices, able to carry out the 6MWT according to the recommendations, and without any known cardiovascular contraindications [23]. Their main sequelae were locomotor disorders due to hemiparesis. The twelve participants with stroke included 10 males and 2 females; their median age was 50.5 years [interquartile range (IQR) 41.25;53.25], their median height was 175 cm [170.0;177.0], and their median weight was 73 kg [62.0;83.0]).

A group of twelve control subjects (8 males and 4 females) was also recruited. Their results were important not only because they allowed a comparison between the two experimental groups, but also because their data ensured that any effects noted were not an artefact of the experimental set-up used in this study, since Bouten’s method has been validated in healthy individuals [23]. Inclusion criteria for the control group were as follows: aged over 18 years and with no known neuromuscular pathologies. Their median age was 29 years [IQR 24.0;33.7], their median height was 177 cm [169.8;177.0], and their median weight was 69 kg [60.0;75.5].

The study was granted ethical approval, all participants provided informed consent for participation, and the study was carried out according to the Helsinki declaration. 

Study design

All participants performed the 6MWT, as recommended by the American Thoracic Society [31], as quickly as possible along a 30 m long corridor that was marked every 2 metres. The distance covered was measured at the end of the test. Participants wore a portable gas exchanger (K4b^2^, COSMED, Rome, Italy) and a three-axis accelerometer (EQO2, Equivital, Cambridge, UK). We chose the 6-min walk test (and its performance criteria including a walk that covers the greatest distance despite the difference in walking speed) due to its common use for functional or cardio-respiratory evaluations.

### 2.2. Procedures

#### 2.2.1. Energy Expenditure: Indirect Calorimetry (EE_META_)

Analysis of the gas expired from each respiratory cycle provided the reference measurement of energy expenditure (EE_META_). The system (K4b^2^, Cosmed, Rome, Italy) was calibrated in the corridor where the test was performed according to standard procedures. 

EE_META_ was calculated when $\dot{V}O_{2}$ kinetics reached a stable state, during the two last minutes of the exercise. The $\dot{V}O_{2}$ values (Kcal·min^−1^) were initially smoothed using a 3-point moving average, then the last 150 s of each 6MWT were averaged. The EE_META_ was then converted to W·kg^−1^.

#### 2.2.2. Energy Expenditure: Accelerometery (${EE}_{ACC})$


Sensor: A lightweight (38 g), compact (78 × 53 × 10 mm), three-axis accelerometer (250 Hz, ±16 g) was positioned between the third and fourth lumbar vertebrae using a custom-made support. This has previously been recommended for the optimal estimation of energy expenditure [18,32,33]. This positioning is considered to be representative of the displacement of the centre of mass in the global coordinate system. A connected chest-strap monitor (EQO2, Equivital, Cambridge, UK) was used to measure heart rate during the 6MWT.Estimation of energy expenditure: Bouten’s method was used to estimate energy expenditure [21]. This method requires accelerometric signal processing in 3 steps:(1)The raw signals were initially filtered using a Butterworth filter (4th order with a 20 Hz cut-off frequency);(2)The absolute values of the signal obtained on the three axes (IAAtot) were then calculated in 30 s periods, and then summed for the duration of the test [21];
$$IAAtot=\int_{i}^{i+30sec}\left|X\right|+\int_{i}^{i+30sec}\left|Y\right|+\int_{i}^{i+30sec}\left|Z\right|$$where *X*, *Y*, and *Z* correspond to the 3 axes of the accelerometer;(3)The following equation was used to obtain predicted EE_ACC_ (W·kg^−1^):



$${EE}_{ACC}=0.104+0.23\times IAAtot$$


### 2.3. Statistical Analysis

The results for the descriptive and interferential statistics were described using the median, and the first and last quartiles (Q1 and Q3). The level of significance was set at *p* ≤ 0.05. The normality of the distribution was verified using a Kolmogorov–Smirnov test.

The results from Bouten’s method and indirect calorimetry were not normally distributed; therefore, we chose to use a Mann–Whitney test to compare the non-homogenous sample with independent samples. The relative agreement between the EE_META_ and EE_ACC_ values of the groups was compared using Spearman’s rank correlations. Absolute agreement was calculated using the Bland–Altman method (± limits of agreement set at 95%) [34].

## 3. Results

The variables measured during the 6MWT are presented in Table 1. There were significant differences between the groups for $\dot{V}O_{2}$ values and distance walked (highest in the control group), but there was no difference in heart rate. 

We observed a significant difference in the median EE_META_ between patients with stroke and the control group. The median EE_META_ was 9.85 [8.18;11.89] W·kg^−1^ in the stroke group and 5.0 [4.56;5.46] W·kg^−1^ in the control group (*p* < 0.0001). For the accelerometric method, the median EE_ACC_ was not significantly different between groups. The EE_ACC_ was 8.2 [7.05;9.56] W·kg^−1^ in the stroke group and 8.57 [7.86;11.24] W·kg^−1^ in the control group (*p* = 0.11) (Table 2). 

The EE_ACC_ and EE_META_ were not significantly correlated in either the control (Spearman’s r = 0.086: *p* = 079) or the stroke groups (Spearman’s r = 0.56: *p* = 0.06) (Table 3).

The Bland–Altman analysis showed large differences between EE_META_ and EE_ACC_ measurements in the stroke group with a mean overestimation of the EE_ACC_ of 1.16 ± 3.70 W·kg^−1^ (*p* = 0.3) relative to the EE_META_ (Figure 1A). In the healthy group, the EE_ACC_ was underestimated by a mean of −2.43 ± 1.45 W·kg^−1^ (Figure 1B).

## 4. Discussion

The purpose of this study was to compare the accuracy of energy expenditure values calculated using an accelerometry signal via Bouten’s regression equation method, with those obtained from the oxygen uptake of indirect calorimetry. The results of this study showed differences between energy expenditure (EE) during a 6MWT calculated using indirect calorimetry (EE_META_) and estimated using Bouten’s method (EE_ACC_) in both healthy volunteers (control) and individuals with stroke. The use of Bouten’s regression equation led to a 17% underestimation in the control group and a 49% overestimation in the calculated energy expenditure in comparison to the gold standard indirect calorimetry results in the stroke group (i.e., EE_ACC_ > EE_META_). 

The first interesting result (Table 2) showed that when EE was calculated using indirect calorimetry, there was a significant difference between the control group and patients with stroke. This difference was consistent with the study by Slawinski, showing that strokes have a lower EE because their walking speed is significantly lower than that of healthy subjects. In that study, the authors also found that the addition of obstacles during a gait test did not affect the $\dot{V}O_{2}$ in patients with stroke [24] as they were already at their $\dot{V}O_{2}$ peak and could not increase their O_2_ consumption further because of their limited gait. Our EE_META_ results agreed with those from that study, namely, that the EE_META_ value of the stroke group participants was half that of the control EE_META_. This difference was mainly due to the difference in the distances covered during performance of the 6MWT: the stroke group covered an average of 339 m, whereas the controls covered, on average, 696 m. Collectively, these results suggest that the reduction in distance covered by patients with stroke was related to an increase in extraneous movements required for movement control and balance in these patients.

The second interesting result concerned the comparison between stroke patients and controls in terms of EE estimated using accelerometery and Bouten’s method. Indeed, these differences could be associated with the fact the Bouten method does not take into account the state of the subject, particularly individual anthropometric characteristics such as the body mass index. However, there was no difference in the EE_ACC_. In other words, the EE_ACC_ was the same for both the stroke patients and controls. These results confirmed the previous hypothesis regarding extraneous movements associated with the locomotion of patients with stroke. In the stroke group, the overestimation of EE_ACC_ using Bouten’s method (compared to gold standard method) was likely due to the individuals’ abnormal segmental kinematics. An increase in vertical oscillations of the pelvis is a common gait anomaly following stroke [28]; it is related to various kinematic anomalies such as knee hyperextension (genu recurvatum) or a stiff gait (lack of knee flexion during swing) [35,36]. The position of the accelerometer just above the pelvis (between L3 and L4) meant that all compensatory movements performed by the subjects as a result of motor and sensory impairments were also recorded. The use of the integral of the unit vector of the accelerometer (IAA_tot_) to calculate EE using Bouten’s method then amplified the EE_ACC_ value. The more the accelerometer moves due to compensating for movements, the higher the amplitude of the accelerometer signals is and the greater the IAA_tot_ is.

The third surprising (Table 3 and Figure 1) result was to find that the control group’s results contrasted with those described by Bouten et al. [23]. Their original paper reported a mean overestimation of EE_ACC_ of 15% in a group of 11 young healthy adults walking at different speeds. However, for a gait speed of 7 km·h^−1^, the EE_ACC_ was overestimated by 8%. By contrast, in the present study, at almost the same gait speed (6.97 ± 0.79 km·h^−1^), Bouten’s method actually underestimated the energy expenditure in the control group by 17%. This contradiction has been observed elsewhere: other studies have also reported both over- and underestimations of EE when using accelerometery and comparing the results to indirect calorimetry in healthy subjects [37]. Indeed, two studies that used an accelerometer device reported opposing results: Bai et al. found an overestimation [38] while Imboden et al. (2018) found an under-estimation [39]. A review of the literature by Jeran et al. in 2016 [40] even shows under- and overestimates of between 3% and 80%. These variations were likely due to differences in the tasks (gait speed, cadence, etc.). There is currently no consensus regarding the level of acceptable errors or whether they relate to under- or overestimations of EE. For strokes, EE_ACC_ overestimated EE by 1.16 ± 3.70 W·kg^−1^. These results confirmed the variability of accelerometric measurements when used to estimate energy expenditure. This measurement variability likely explains the lack of correlation observed between the two measurement methods.

The present results associated with those of previous studies show that there is currently no consensus regarding the level of acceptable errors or whether they relate to under- or overestimations of EE. The variety of EE_ACC_ results obtained by different research groups suggests that it is important to be aware of the limitations in the use of accelerometers. We recommend that, in order to take advantage of the convenience of accelerometer measurements, healthcare practitioners should produce their own reference data within their own setting and in patients with different pathologies using both indirect calorimetry and accelerometery in order to make informed interpretations of the accelerometery data. 

The present study was different to other studies of EE_ACC_ with regards to two methodological aspects: (1) the choice of accelerometer signal processing method and (2) the positioning of the sensor. In terms of the first point, signal processing using the root-mean square has been largely replaced by count per minute [41]. Nevertheless, there is currently no accepted consensus in the literature regarding threshold values for activity detection. This may, at least in part, be due to inter-individual variations caused by variables such as age or existing medical conditions. For example, it has been reported that it is difficult to calculate EE_ACC_ in older patients when using gait thresholds taken from younger adults, as older people have a naturally wider range of inappropriate movements compared to younger adults which led to an unreliable detection of EE in the older population [17].

With regards to sensor position, Compagnat et al. [42] found a mean difference in the predicted energy expenditure between 3% and 58% in patients with hemiparesis when the sensor was positioned on the wrist rather than the pelvis. However, Bouten et al. (1997) recommended positioning the sensor between L3 and L4 [18] in order to quantify movement of the centre of mass, and this position has been used in many studies [19,21,25,36,37]. We think that it seems more logical to place the sensor around the pelvis if the aim is to record compensatory gait movements, and to objectify the patient’s progress during rehabilitation. Finally, recent works demonstrated [43] that the choice of the oxygen cost prediction equation can greatly improve the estimation of stroke patients’ daily energy expenditure. 

The main limitation of this study was the inclusion of patients with diverse gait patterns. Unfortunately, there were too few patients with each type of gait pattern to determine the effects of different compensatory movements and to refine the prediction equation accordingly. On the other hand, our sample was small and did not walk at the same speed. We also observed a mismatch between gender and age.

## 5. Conclusions

In conclusion, therefore, the results from this study suggest that the 1993 Bouten method [22] does have the potential to be of considerable practical value for quantifying rehabilitation-induced changes in gait (improvement in gait and reduction in compensatory movement acting on the IAA_tot_). In addition to being cheaper and more accessible than indirect calorimetry, Bouten’s method to assess energy expenditure using an accelerometer also accounts for compensatory lower limb movements that occur as part of a pathological gait. However, we should also stress that this method is, at present, unvalidated in the wider research community and is not always predictable in terms of how EE_ACC_ results vary in comparison to EE_META_, even in the same populations. Therefore, this tool is not yet suitable for use as a stand-alone measurement in routine practice for the assessment of any patients with stroke-related hemiparesis and impaired gait. Further investigations are required to ensure that the necessary corrective coefficients are known for different patient groups and pathologies in order to ensure the accuracy, reliability, and reproducibility of EE_ACC_ values for the different combinations of patient demographics and pathologies.

## Figures and Tables

**Figure 1 sensors-23-07177-f001:**
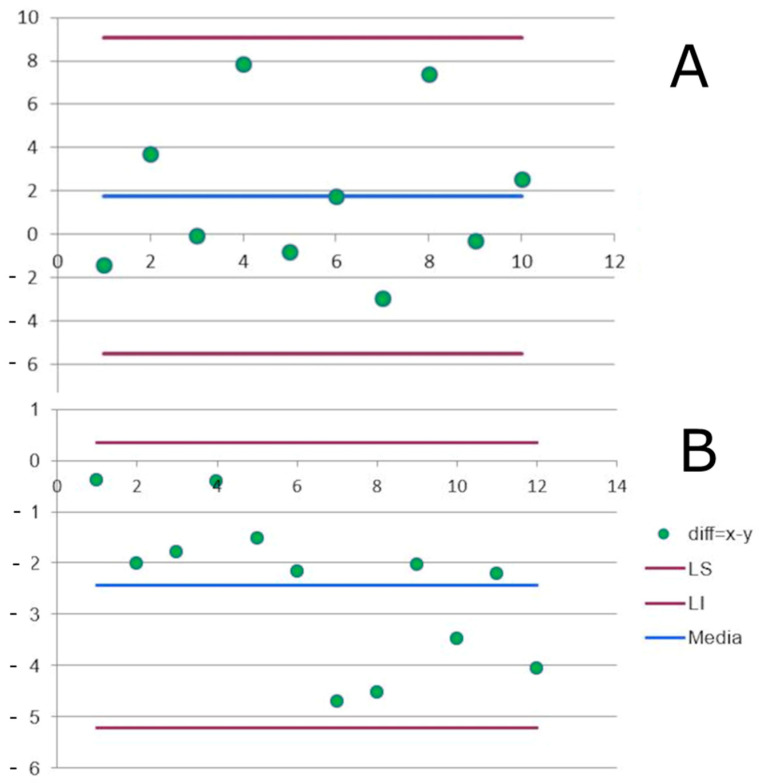
Bland–Altman stroke (**A**) and healthy participant. UL (**B**) between EE_META_ and EE_ACC_.

**Table 1 sensors-23-07177-t001:** Variables measured during 6MWT.

	Control Group			Patient with Stroke			Mann–Whitney
	Median	Q1	Q3	Median	Q1	Q3	
HR (bpm)	140.0	98.1	143.7	116.0	90.1	126.5	*p* = 0.08
$\dot{V}O_{2}$ (mL·min^−1^·kg^−1^)	28.65	23.35	33.83	13.55	12.63	15.8	*p* = 0.0001
Distance (m)	686.5	660.0	729.7	341.0	310.0	442.0	*p* = 0.0001

**Table 2 sensors-23-07177-t002:** Comparison of EE measurements between methods (EE_ACC_ and EE_META_). * *p*-value < 0.5.

		Median	Q1	Q3	Mann–Whitney
EE_META_ (W·kg^−1^)	Control group	9.85	8.18	11.89	*p* < 0.0001 *
	Patient with stroke	5.0	4.56	5.46	
EE_Acc_ (W·kg^−1^)	Control group	8.57	7.86	11.24	*p* = 0.11
	Patient with stroke	8.2	7.05	9.56	

**Table 3 sensors-23-07177-t003:** Correlation between energy expenditure measured using K4b^2^ and the accelerometric method.

		Median	Q1	Q3	Correlation Coefficient
Control group	EE_META_ (W·kg^−1^)	9.85	8.18	11.89	r = 0.09; *p* = 0.79
	EE_Acc_ (W·kg^−1^)	8.57	7.86	11.24	
Patient with stroke	EE_META_ (W·kg^−1^)	5.0	4.56	5.46	r = 0.56; *p* = 0.06
	EE_Acc_ (W·kg^−1^)	8.2	7.05	9.56	

## Data Availability

Not applicable.

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
