# Peer review of "Does Accelerometry at the Centre of Mass Accurately Predict the Gait Energy Expenditure in Patients with Hemiparesis?"

_sensors, 2023, doi:10.3390/s23167177_

Round 1

Reviewer 1 Report

1. The research method is not well described. It would be expedient to add to Section 2 a schematic description of the Bouten's method, including the coordinate frame, positions of the sensor relative to the center of mass, and the corresponding equations of motion.

2. There are no important parameters of the sensor in the manuscript, such as bias, dynamic range. Only weight and size characteristics are given.

3. It is not clear whether the influence of external factors and noise was taken into account.

Author Response

Reviewer 1

Answer: We thank the reviewer for the work on our paper. Please find our detailed answer

The research method is not well described. It would be expedient to add to Section 2 a schematic description of the Bouten's method, including the coordinate frame, positions of the sensor relative to the center of mass, and the corresponding equations of motion.

Answer : We have redesigned this part, lines 142-156, so that the calculation steps such as the sensor information are more readable. We have also detailed the calculation of IAAtot proposed by Bouten et al. [21].

There are no important parameters of the sensor in the manuscript, such as bias, dynamic range. Only weight and size characteristics are given.

Answer: Thank you for your comment. We have added the acquisition frequency and the range line 143.

It is not clear whether the influence of external factors and noise was taken into account.

Answer: we understand the remark and we attach importance to the calculation steps being as clear as possible so that other teams can reproduce the method. For this we have indicated the filtering parameters line 151 and detailed the IAAtot calculation line 155.

Reviewer 2 Report

The authors of the study entitled "A triaxial accelerometer and portable data processing unit for the assessment of daily physical activity" propose to study the possibility of evaluating metabolic energy expenditure using an accelerometer with the Bouten’s equation. It is here analysing a healthy population and post-stroke. 

This work poses a simple and relevant question of transferring a calibrated method for a healthy population (Bouten’s equation) to a pathological population.

Although the document is well structured and organized, it is necessary to make some corrections and additions before accepting publication.

1)The article refers to an old bibliography on the link between accelerometer measurements and energy expenditure (EE) assessment. In the past 3 years, there has been a lot of research related to this estimate that provides metrics for quantifying activity levels or metabolic energy. It is also necessary in what way the Bouten’s equation is more relevant here than any other used.

2)There is a bias in the use of indirect calorimetry. At the beginning of the trial (about the first two minutes), the activity is anaerobic, and the system cannot measure energy expenditure indirectly. Perhaps this is the reason for the observed underestimation ?

3)The equation on line 149 on page 4 should be revised or the text on it corrected, as there is an inconsistency between the two. It is also surprising that anthropometric data of the subject are not taken into account - this may also be a subject for discussion.

4)The discussion should emphasize more than the fact that the equation used here does not take into account the state of the subject at all; this is why the authors find there is no difference (line 221 page 7)

5)The most disturbing result is that the Bouten’s equation does not give satisfactory results even with healthy subjects. This is all the more worrisome since the authors conclude that this suggests that the measurement of the accelerometer is too variable to allow quantification of the EE (line 249 page7)  here, the question of the validity of all previous studies is asked. The authors must deepen their reflection here.

6)At the end of the discussion, the authors propose to place the sensor on the pelvis, this can be more argued. Moreover, there are numerous publications that have evaluated the impact of the position of the sensor, but also of the metric and the post-processing methods. This needs to be discussed in depth.

Author Response

Reviewer 2

The authors of the study entitled "A triaxial accelerometer and portable data processing unit for the assessment of daily physical activity" propose to study the possibility of evaluating metabolic energy expenditure using an accelerometer with the Bouten’s equation. It is here analysing a healthy population and post-stroke. This work poses a simple and relevant question of transferring a calibrated method for a healthy population (Bouten’s equation) to a pathological population. Although the document is well structured and organized, it is necessary to make some corrections and additions before accepting publication. 

Answer: We thank the reviewer for his comments and his work made to improve the present document. We tried to give a detailed answer to his questions.

1)The article refers to an old bibliography on the link between accelerometer measurements and energy expenditure (EE) assessment. In the past 3 years, there has been a lot of research related to this estimate that provides metrics for quantifying activity levels or metabolic energy. It is also necessary in what way the Bouten’s equation is more relevant here than any other used.

Answer: Thank you for your comment.

Recently, some authors have been interested in predicting energy expenditure in stroke survivors (https://doi.org/10.1177/2047487317738593). However, they use predictions made directly by industrial software. For example, the Actigraph system uses the freedson equation. The latter uses the number of counts as an index in the measurement. However, this information is protected by an industrial secret (https://doi:101007/s00421-006-0307-5). The aim of this work is to challenge the equations available, including BOUTEN's equation.

We have added the following sentence (line 39) to make it easier for the reader to understand:

“However, the methods used to predict energy expenditure are subject to industrial secrets, as in the study by Mandigout and Compagnat (line 40), where the prediction made by the accelerometer is protected by an industrial secret.”

2)There is a bias in the use of indirect calorimetry. At the beginning of the trial (about the first two minutes), the activity is anaerobic, and the system cannot measure energy expenditure indirectly. Perhaps this is the reason for the observed underestimation? 

Answer: Thank you for this remark, indeed you are right, that’s why we have made the measurement during the last 150s of the exercise.

This is now indicated line 137: “EEMETA was calculated when kinetics reached a stable state, during the two last minute of the exercise”

And line 139:” then the last 150 seconds of each 6MWT were averaged”

3)The equation on line 149 on page 4 should be revised or the text on it corrected, as there is an inconsistency between the two. It is also surprising that anthropometric data of the subject are not taken into account - this may also be a subject for discussion. 

Answer: Thank you for your vigilance. We used the term IAAtot as mentioned in the methodology proposed by bouten's reference article.

 Bouten et al., 1994 : Integrals of the absolute value of accelerometer output from x, y, z directions were obtained by rectification and integration of the signals over the 30-s time interval, resulting in the variables IAAx, IAAy and IAAz. the sum of these variables was calculated to get IAAtot.

4) The discussion should emphasize more than the fact that the equation used here does not take into account the state of the subject at all; this is why the authors find there is no difference (line 221) 

Answer: Thank you for your comment, which is absolutely correct. We have added the following sentence on line 231.

Indeed these differences could be associated to the fact the Bouten method does not take into account the state of the subject, and particularly the individual anthropometric caracteristics such as the body mass index.

5)The most disturbing result is that the Bouten’s equation does not give satisfactory results even with healthy subjects. This is all the more worrisome since the authors conclude that this suggests that the measurement of the accelerometer is too variable to allow quantification of the EE (line 249)  here, the question of the validity of all previous studies is asked. The authors must deepen their reflection here. 

Answer: Thank you for your comment, which we share.

These results are however recurrent in the literature with, for example, over and underestimates of 3 to 80% highlighted by a review of the literature by Jeran et al. in 2016.

We have added the following sentence on line 258.

A review of the literature by Jeran et al. in 2016 [40] even shows under- and over-estimates of between 3% and 80%.

6)At the end of the discussion, the authors propose to place the sensor on the pelvis, this can be more argued. Moreover, there are numerous publications that have evaluated the impact of the position of the sensor, but also of the metric and the post-processing methods. This needs to be discussed in depth.

Answer: Thank you for your reply. The sensor was positioned as close as possible to the center of mass (between L3 and L4) as indicated in Bouten's methodology.

However, we know that positioning is a determining factor in measuring energy expenditure, as this article shows:10.1016/j.jstrokecerebrovasdis.2022.106397

Reviewer 3 Report

Dear authors, this is a very interesting work, nicely written, in an important field of research, based on the epidemiology of the disease. However, some recommendations are suggested.

First, would you please position your work in terms of novelty, comparing to the available literature? Because the similarity of your work and previously published literature is considerable. For example, this work: https://academic.oup.com/eurjpc/article/24/18/2009/5926653

And many other studies in the field have been developed, that you shoud describe, to position your study.

Secondly, would you please provide the ethical approval committee number and the reference to the institution which you have applied?

Please consider evaluating the normality of your work using shapiro wilk test, that is more adequate for small samples.

Author Response

Reviewer 3

Dear authors, this is a very interesting work, nicely written, in an important field of research, based on the epidemiology of the disease. However, some recommendations are suggested.

Answer: We greatly thanks the reviwer for his positive comments about the present Work. We tried to improve the present work thought a detailed answer to yours questions.

First, would you please position your work in terms of novelty, comparing to the available literature? Because the similarity of your work and previously published literature is considerable. For example, this work: https://academic.oup.com/eurjpc/article/24/18/2009/5926653

Answer: As suggested by the reviewer 2 we have some recent literature in order to improve the rationale of the present study.

And many other studies in the field have been developed, that you shoud describe, to position your study.

Answer: As suggested by the reviewer 2 we have some recent literature in order to improve the rationale of the present study. You can read now line 38.

Secondly, would you please provide the ethical approval committee number and the reference to the institution which you have applied?

Answer: The number of the study: France #2011-A01487-34 - NCT01807247; update 2015-01-04.

Please consider evaluating the normality of your work using shapiro wilk test, that is more adequate for small samples.

Answer: Independently, of the sample size (small sample) the studied parameters did not respect the normality law, thus we made non-parametric statistical test (Mann and Whitney). This is precised in the statistical paragraph.

Round 2

Reviewer 2 Report

Additional information well completed the paper.